# Bioassay-Guided Discovery of Potential Partial Extracts with Cytotoxic Effects on Liver Cancer Cells from Vietnamese Medicinal Herbs

Hien Minh Nguyen [1,2,*], Nhi Yen Thi Nguyen [2,3], Nghia Trong Ngoc Chau [1,2], Anh Bao Thi Nguyen [1,2], Van Kieu Thi Tran [2,4], Viet Hoang [2,4], Tri Minh Le [1,2], Hui-Chun Wang [5,6] and Chia-Hung Yen [5,6]

1. School of Medicine, Vietnam National University Ho Chi Minh City, Quarter 6, Linh Trung Ward, Thu Duc Distrcut, Ho Chi Minh City 700000, Vietnam; cntnghia.d2017@medvnu.edu.vn (N.T.N.C.); ntbanh@medvnu.edu.vn (A.B.T.N.); lmtri@medvnu.edu.vn (T.M.L.)
2. Vietnam National University Ho Chi Minh City, Quarter 6, Linh Trung Ward, Thu Duc District, Ho Chi Minh City 700000, Vietnam; nhi.nguyen0610@hcmut.edu.vn
3. Ho Chi Minh City University of Technology, 268 Ly Thuong Kiet Street, Ward 14, District 10, Ho Chi Minh City 700000, Vietnam
4. University of Science, Vietnam National University Ho Chi Minh City, Quarter 6, Linh Trung Ward, Thu Duc District, Ho Chi Minh City 700000, Vietnam; ttkvan.hcmus@gmail.com (V.K.T.T.); hviet@hcmus.edu.vn (V.H.)
5. Graduate Institute of Natural Products, College of Pharmacy, Kaohsiung Medical University, No.100, Shih-Chuan 1st Road, Sanmin District, Kaohsiung City 80708, Taiwan; wanghc@kmu.edu.tw (H.-C.W.); chyen@kmu.edu.tw (C.-H.Y.)
6. National Natural Product Libraries and High-Throughput Screening Core Facility, Kaohsiung Medical University, No.100, Shih-Chuan 1st Road, Sanmin District, Kaohsiung City 80708, Taiwan
* Correspondence: nmhien@medvnu.edu.vn; Tel.: +84-373-696-894

**Abstract:** Hepatocellular carcinoma (HCC) is the most frequent type of primary liver cancer and is the leading cause of cancer mortality in Vietnam. Our study aims to discover the partial extracts with the potential cytotoxic effects on HCC cells from the different parts of 24 Vietnamese medicinal plants traditionally used in liver cancer treatment. Out of 52 crude methanol extracts, we found that *Luvunga scandens* leaves, *Hyptis suaveolens* roots, and *Solanum torvum* leaves showed the notable cytotoxic effects against HCC cells. After that, we carried out partial extract of the three methanol extracts with ethyl acetate, water, *n*-hexane, and 90% methanol. The cytotoxic activity on Huh-7 HCC cells, antioxidant capacity, and total flavonoids content (TFC) of each partial extraction were determined. Methanol, ethyl acetate, and 90% methanol extracts showed moderate to strong cytotoxicity activity against Huh-7 HCC cells. Notably, the ethyl acetate and 90% methanol extract from *H. suaveolens* roots with high TFC values and strong antioxidant capacity could be promising sources of novel therapeutic modalities for HCC treatment. For the leaves of *L. scandens* and *S. torvum*, the ethyl acetate extract showed high TFC value and promising anti-HCC activity, therefore recommended further studies.

**Keywords:** *Luvunga scandens*; *Hyptis suaveolens*; *Solanum torvum*; Huh-7 cells; cytotoxic activity; antioxidant; flavonoids; hepatocellular carcinoma

## 1. Introduction

Liver cancer has become a warning problem with 905,677 new cases, 830,180 new deaths [1], and was the leading cause of cancer-related death in Vietnam recorded in 2020 [2]. The main two forms of primary liver cancer are hepatocellular carcinoma (HCC) and intrahepatic cholangiocarcinoma [3]. HCC is the most common type of primary liver cancer, comprising 75–85% of cases [1]. The HCC prognosis is frequently poor as most HCC patients are detected at the late stage of liver cancer; thus, the mortality rate is exceptionally high and approximately equal to the incidence rate [4–6].

There are several treatment options for HCC patients based on the stage of HCC. For early-stage of HCC, surgical resection is suggested because of safety and effectiveness, even in elderly patients [7,8]. As well as surgical resection, chemotherapy is also regularly performed in the HCC treatment. However, further problems from using chemotherapy drugs such as toxicity, drug inefficacy, and some severe side effects for long-time treatment led to insignificant effectiveness in improving outcomes [9]. An unconventional and long-term method to treat chronic diseases, including HCC, is the use of medicinal herbs because of a certain number of benefits such as high bioactivity, less unexpected effects, and less toxicity to normal cells [10,11]. There are 62 compounds among 247 anticancer drugs approved from January 1981 to September 2019 that are natural products or derived from a natural product [12]. In addition, more than half of the medicines approved by the U.S. Food and Drug Administration for the treatment of cancer are natural products or plant-derived natural products [13]. Consequently, the use of biologically active substances from medicinal herbs in supporting the treatment and prevention of cancer is a trend that deserves attention.

Vietnam has plentiful medicinal plant resources with more than 4700 species used as a medicine, which is advantageous for screening herbals acting on HCC cells [14]. For a long time, traditional herbal remedies were used for the treatment of malignant and chronic diseases, including ascites, cirrhosis, and liver diseases [14–17]. However, the current use of medicinal plants in Vietnam is based chiefly on folk experience or traditional documents such as Medicinal plants in An Giang Province (1991) [15], Plants and animals used as medicine in Vietnam (2004) [16], Vietnamese Medicinal Plants and Remedies (2004) [17], and Dictionary of Vietnamese Medicinal Plants (2012) [14]. The most significant advantage of screening the truly beneficial herbs is improving treatment effect and saving time as well as money for patients. Additionally, the screening investigation also allows researchers to discover new bioactive natural products. Therefore, studying and expanding the screening experiments for nations with plenty of medicinal plants like Vietnam is essential. Typically, when plant extraction was shown to be biologically active, a bioassay-guided partial extraction was carried out to find the potential extract for isolation, identification compounds, and drugs development [12,18].

In the present study, we conducted a cytotoxic screening against three liver cancer cell lines Huh-7, Hep3B, and HepG2, with 52 crude methanol extracts prepared from different parts of 24 medicinal herbs. These medicinal plants were recommended by the Tinh Bien District Orientally Traditional Medicine Association, An Giang Province, Vietnam based on folk experience, traditional remedies, and the Vietnamese traditional documents for the treatment of liver cancer and ascites for a long time [14–17]. Three hit crude extracts from *Luvunga scandens* leaves, *Hyptis suaveolens* roots, and *Solanum torvum* leaves were selected for partial extraction to yield four additional partial extracts from each crude extract. Here, the total flavonoids content (TFC), the antioxidant activity, and the potency of cytotoxic effect against Huh-7 HCC cells of these extracts were determined.

## 2. Materials and Methods

### 2.1. Chemicals

Ethyl acetate, *n*-hexane, aluminum chloride hexahydrate, sodium nitrite, and dimethyl sulfoxide (DMSO) were purchased from Merck KgaA (Darmstadt, Germany), rutin, Dulbecco's modified Eagle's medium (DMEM), and 3-(4,5-dimethylthiazol-2-yl)-2,5-diphenyltetrazolium bromide (MTT) were purchased from Sigma-Aldrich (St. Louis, MO, USA), whereas methanol, resazurin, and 1,1-diphenyl-2-picrylhydrazyl (DPPH) were purchased from Prolabo (Paris, France), Cayman Chemical Company (Ann Arbor, MI, USA), and Tokyo Chemical Industry (Tokyo, Japan), respectively.

### 2.2. Instrumentation

Hitachi U-3900 Spectrometer (Hitachi, Tokyo, Japan), Rotary Evaporator RE300 (Stuart, London, UK), Synergy HT Multi-Mode Reader (Bio-Tek Instruments Inc., Winooski, VT,

USA), Shimadzu UV-1800 Spectrophotometer (Shimadzu, Kyoto, Japan), Microplate Reader Thermo Multiskan Ascent (Thermo Fisher Scientific, Waltham, MA, USA), NuAire $CO_2$ Incubators (NuAire, Plymouth, MN, USA).

### 2.3. Collecting, Preparing, and Extracting Plant Materials

All plant materials were collected in Tinh Bien District, An Giang Province, Vietnam in October 2020 and authenticated by Dr. Hoang Viet, Department of Ecology-Evolutionary Biology, University of Science, Vietnam National University Ho Chi Minh City (Table 1). These fresh samples were washed, drained, cut into small pieces, and completely dried at 50 °C before being ground and sieved using a strainer with 1 mm mesh size pore. Each plant powder (20.0 g) was soaked in methanol (160 mL) for 3 days, filtered with Newstar 102 filter paper, and the solvent was removed using a vacuum evaporator to yield the crude methanol extract.

### 2.4. Cell Cultures

Huh-7, Hep3B, and HepG2 HCC cell lines were kindly provided by Professor Yan-Hwa Wu Lee (Institute of Biochemistry and Molecular Biology, School of Life Sciences, National Yang-Ming University). All three cell lines were cultured in complete DMEM. The DMEM medium included 10% fetal bovine serum (Gibco), penicillin (100 U/mL), streptomycin (100 μg/mL), nonessential amino acids (0.1 mM), and L-glutamine (2 mM). The cells were maintained at 37 °C and an atmosphere of 5% $CO_2$.

### 2.5. Cytotoxicity Screening ASSAY

The crude methanol extracts were screened for cytotoxicity effect on Huh-7, Hep3B, and HepG2 HCC cell lines by using MTT assay. Briefly, the cells ($1 \times 10^4$) were seeded in a 96-well plate with the cell culture medium (100 μL) and treated with the plant extracts for 48 h. The concentrations of plant extracts were 100 μg/mL. DMSO at a concentration of 1% was used as a control group. After incubation, the cell cultures were changed to MTT solution with the final concentration was 0.5 mg/mL and continue incubated for 4 h. Formazan crystals were dissolved in DMSO, and the optical density was measured at 550 nm using the Microplate Reader Thermo Multiskan Ascent [19,20].

The cell viability was calculated using the equation:

$$\% \text{ Cytotoxicity effect} = (OD_{control} - OD_{test})/OD_{control} \times 100\%, \tag{1}$$

The experiment was performed in three times and all data are presented as mean $\pm$ SD.

### 2.6. Preparation of Partial Extracts

The crude methanol extracts of designated medicinal herbs were partitioned by a two-phase solvent system ethyl acetate: water (4:3, *v/v*), with solid: liquid ratio (1:4, g/L) in the separating funnel. Shook vigorously and allowed for the solvent layer to separate completely for 1 min. After phase separation, the water layer was then collected, followed by the partial extraction process, which was repeated four times with the same ratio to obtain ethyl acetate layer and water layer separately. All ethyl acetate layers and water layers were combined, respectively, and then the solvents were removed using a vacuum evaporator to obtain ethyl acetate extract and water extract. Ethyl acetate extract was then partitioned in a two-phase solvent system consisting of *n*-hexane: 90% methanol (1:1, *v/v*). This step was performed the same as the first partition step to yield *n*-hexane phase and 90% methanol phase. The solvent removal was done by using a vacuum evaporator to obtain *n*-hexane extract and 90% methanol extract. The partial extracts were then stored at 4 °C. The partial extraction process is shown in Figure 1.

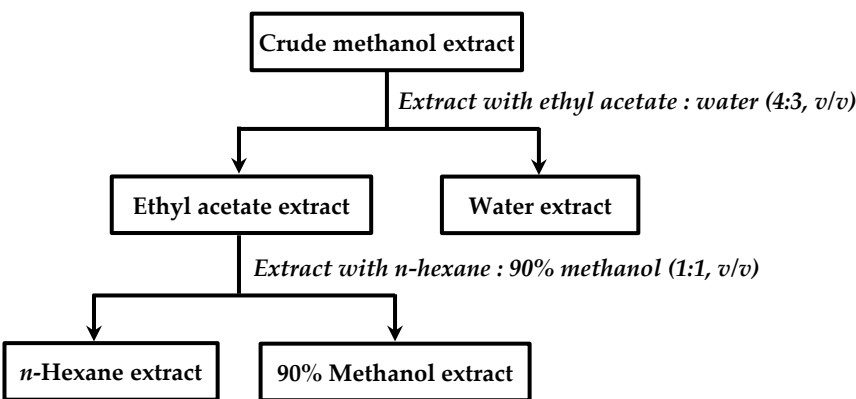

**Figure 1.** Partial extraction procedure.

### 2.7. Total Flavonoids Content (TFC)

Total flavonoids content was determined by using the aluminum-flavonoid complex colorimetric method [21,22]. Exact amounts (0.2, 0.4, 0.8, 1.2, 1.6, 2.0 mL) of the standard solution of rutin (250 μg/mL) were added into 10 mL volumetric flasks. The standard solutions were then mixed with sodium nitrite solution (5%, 0.3 mL). After 5 min, aluminum chloride solution (2%, 0.3 mL) was added, rested for 6 min, and mixed with sodium hydroxide solution (1 M, 3.0 mL). The volume was adjusted to 10 mL with diluted water and allowed to stand at room temperature for 10 min. The amount of aluminum chloride solution (2%) was substituted with the same amount of water in the blank. The absorbance was measured at 510 nm against the blank using Hitachi U-3900 Spectrophotometer. All analyses were performed in triplicate.

The same procedure was applied for all crude extracts and partial extracts. TFC was calculated as rutin equivalents per gram of dry weight of extract (mg RE/g extract) based on the formula:

$$TFC = (a \times V)/m, \tag{2}$$

In which:
TFC: total flavonoids content (mg RE/g extract).
a: value from the calibration curve of rutin standard (μg/mL).
V: volume of total solution (L).
m: mass extract in V (g).

### 2.8. DPPH Free Radical Scavenging Assay

The antioxidant activity was evaluated following a previously described DPPH free radical scavenging capacity method [23]. DPPH can produce stable free radicals in methanol solution with specific purple color and react with antioxidants yielding a yellow complex. In brief, a serial dilution (the concentration range between 5 and 650 μg/mL) of the extract were mixed with a methanol solution of DPPH (0.2 mM, 1.1 mL) and adjusted the volume to 3 mL with methanol. The mixtures were stored in the dark at room temperature for 15 min. The control sample was prepared at the same conditions without the extracts. After that, the absorbance of the samples was measured at 516 nm corresponding to the maximum absorbance peak of DPPH using the Shimadzu UV-1800 UV/Visible Scanning Spectrophotometer with methanol as a blank. Each experiment was performed in triplicate.

The percentage of DPPH radical scavenging was estimated using the equation:

$$SC\% = (As - Ae)/As \times 100\%, \tag{3}$$

In which:
SC%: percentage of DPPH free radical scavenging effect.
As: absorbance of the standard.

Ae: absorbance of the extractives.

The half maximal inhibitory concentration ($IC_{50}$) value of the extracts was calculated at the value that inhibits 50% free radical DPPH and is calculated by interpolation.

### 2.9. Determination of 50% Cytotoxic Concentration ($CC_{50}$)

The Huh-7 HCC cells were cultured in complete DMEM as described by Li et al. [24]. The Huh-7 HCC cells were seeded in a 96-well plate with cell culture medium (100 µL) and treated with the plant extracts for 72 h. The concentrations of plant extracts were ranged from 6.25–100 µg/mL. Resazurin solution was added and incubated for an additional 4 h at 37 °C and 5% $CO_2$. After incubating, resazurin fluorescence (ex/em: 530 nm/590 nm) was measured from the culture supernatant using a Synergy HT Multi-Mode Reader to determine cell viability. The % cell viability values were plotted and used to determine $CC_{50}$. Each experiment was repeated in triplicate.

### 2.10. Statistical Analysis

Statistical analysis was performed using Microsoft Excel 2016. The results of $CC_{50}$ were processed using GraphPad Prism 9 statistical software. All data were presented as means $\pm$ SD for three replications for each experiment.

## 3. Results

### 3.1. Cytotoxicity Screening

The results of cytotoxicity screening on the three HCC cell lines with 52 extracts prepared from different parts (roots, leaves, stems, fruits, and rhizomes) of 24 medicinal herbs were presented in Table 1. Cytotoxicity effect values (%) showed the cytotoxicity of crude methanol extracts at a concentration of 100 µg/mL; the higher the value, the more cytotoxic activity. Most of the crude extracts did not cause significant cytotoxicity to these three cells. The crude extracts from leaves of *Luvunga scandens* (Roxb.) Buch.-Ham. ex Wight & Arn. (*L. scandens* leaves), roots of *Hyptis suaveolens* (L.) Poit. (*H. suaveolens* roots), and leaves of *Solanum torvum* Sw. (*S. torvum* leaves) exhibited cytotoxicity higher than 50% against at least one HCC cell line and were selected for further studies. Among them, the extracts of *H. suaveolens* roots were found to be the most potent extract that induced 73%~85% cytotoxicity in all three HCC cell lines. The extracts of *L. scandens* leaves cause 54% and 73% cytotoxicity in Hep3B and HepG2 HCC cells, respectively. The extract of *S. torvum* leaves showed notable cytotoxicity only to Hep3B HCC cells with a cytotoxic effect of around 77%.

### 3.2. Evaluation of TFC, Antioxidant Activity, and $CC_{50}$ of the Extracts from Luvunga scandens Leaves

The total flavonoids content of the extracts from *L. scandens* leaves are shown in Figure 2a and Tables S1–S3. The flavonoid compounds were enriched in ethyl acetate layer (233.46 $\pm$ 17.34 mg RE/g extract), and in *n*-hexane layer (354.89 $\pm$ 32.55 mg RE/g extract). As presented in Figure 2b, the extracts from leaves of *L. scandens* did not show antioxidant activity with $IC_{50}$ values > 250 µg/mL [49]. The methanol, ethyl acetate, and 90% methanol extracts from *L. scandens* leaves exhibited moderate cytotoxicity activity with $CC_{50}$ 36.5 $\pm$ 2.9, 38.5 $\pm$ 2.1, 51.2 $\pm$ 1.5 µg/mL, respectively.

### 3.3. Evaluation of TFC, Antioxidant Activity, and $CC_{50}$ of the Extracts from Hyptis suaveolens Roots

The total flavonoids content, antioxidant activity, and cytotoxic effect of the extracts from *H. suaveolens* roots were described in Figure 3 and Tables S1–S3, respectively.

The total flavonoids content of *H. suaveolens* roots was highest in the ethyl acetate extract with TFC value was 312.23 $\pm$ 22.82 mg RE/g extract, followed by those of 90% methanol extract and water extract with 259.75 $\pm$ 6.96 mg RE/g extract and 249.67 $\pm$ 23.71 mg RE/g extract, respectively (Figure 3a, Table S1).

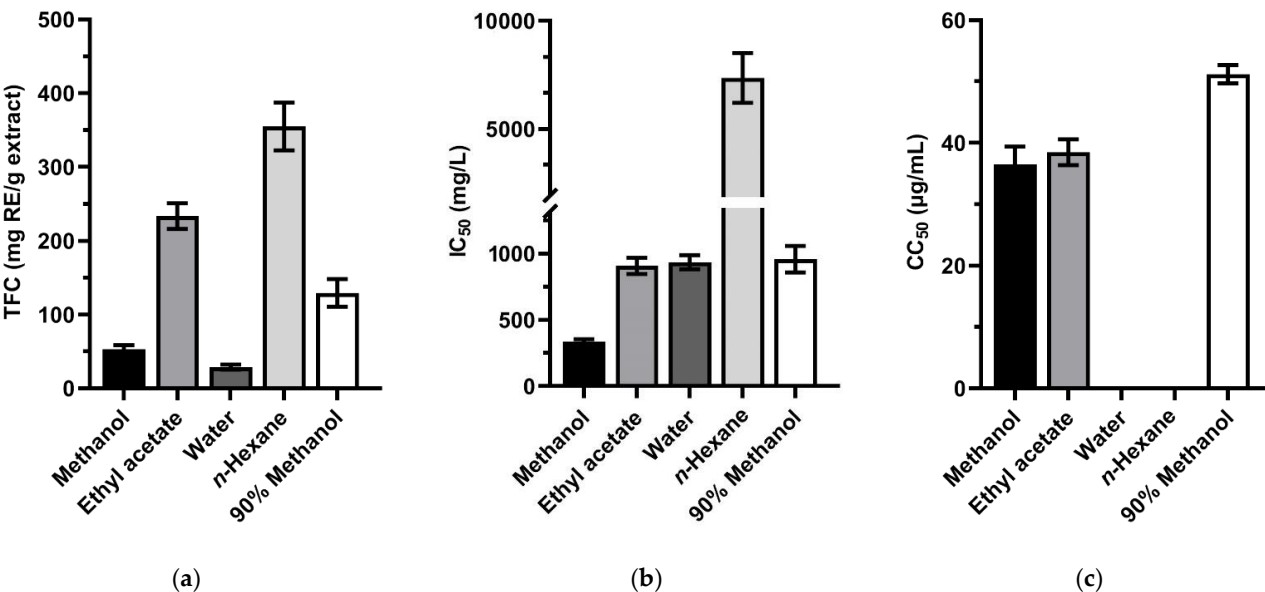

**Figure 2.** (**a**) Total flavonoids content; (**b**) the antioxidant activity in DPPH assay; (**c**) the cytotoxicity effect on Huh-7 HCC cells of *Luvunga scandens* leaves extracting in methanol, ethyl acetate, water, *n*-hexane, and 90% methanol. For panel (**c**), no bar is shown for the extracts that had a 50% cytotoxic concentration of more than 100 μg/mL. Values are presented as mean ± SD (*n* = 3). Error bars show standard deviation.

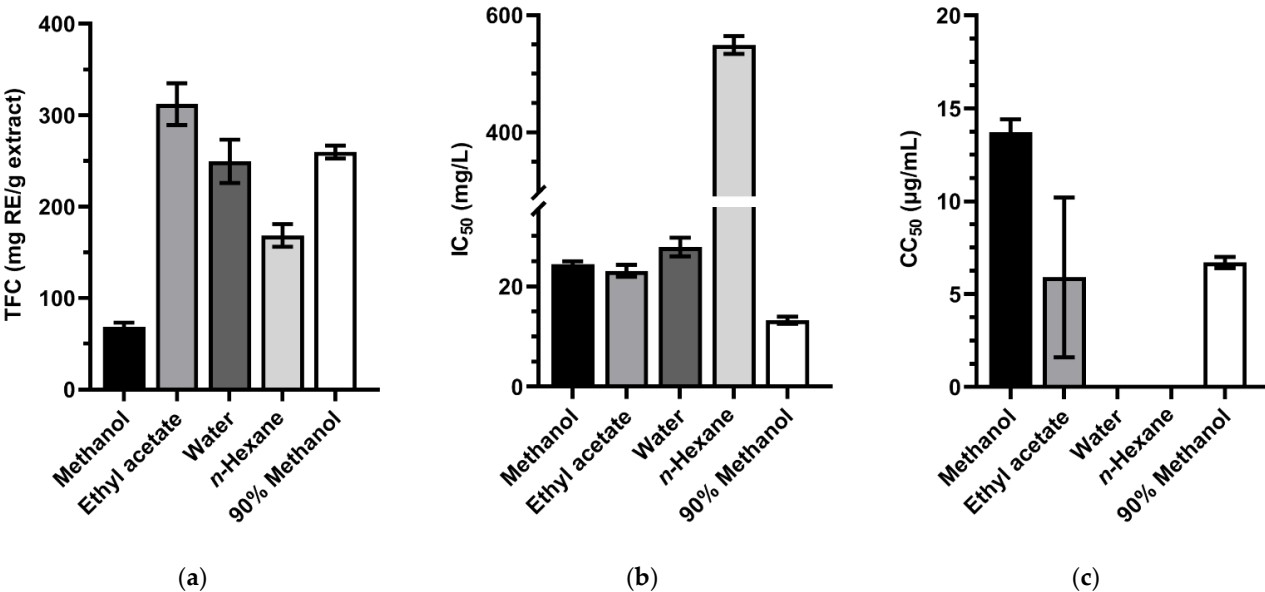

**Figure 3.** (**a**) Total flavonoids content, (**b**) the antioxidant activity in DPPH assay and (**c**) the cytotoxicity effect on Huh-7 HCC cells of *Hyptis suaveolens* leaves extracting in methanol, ethyl acetate, water, *n*-hexane, and 90% methanol. For panel (**c**), no bar is shown for the extracts that had a 50% cytotoxic concentration of more than 100 μg/mL. Values are presented as mean ± SD (*n* = 3). Error bars show standard deviation.

Except for *n*-hexane extract, the remaining extracts from *H. suaveolens* roots exhibited strong radical scavenging activity [49]. The 90% methanol extract was the most potent extract with $IC_{50}$ of $13.27 \pm 0.68$ μg/mL (Figure 3b, Table S2).

In Figure 3c and Table S3, the methanol, ethyl acetate, and 90% methanol extracts from *H. suaveolens* roots expressed a strong cytotoxicity effect with $CC_{50}$ $13.7 \pm 0.7$, $5.9 \pm 4.3$, and $6.7 \pm 0.3$ μg/mL, respectively.

**Table 1.** In vitro screening of 52 crude methanol extracts from different parts of 24 medicinal herbs for cytotoxicity effect on Huh-7, Hep3B, and HepG2 HCC cells using MTT assay.

| No. | Scientific Name | Plant Family | Traditional Use | Plant Part | Cytotoxicity Effect (%) [1] | | |
|---|---|---|---|---|---|---|---|
| | | | | | Huh-7 Cells | Hep3B Cells | HepG2 Cells |
| 1 | *Piper sarmentosum* Roxb. | Piperaceae | Rheumatism arthralgia, rheumatic bone pain, wet joint pain and traumatic injury treatment [16,17]. Abdominal distension, gastricism, stomachache, toothache, headache [16]. Breast cancer [25], cervical carcinoma, and colon adenocarcinoma [26]. | Roots | 19.70 ± 5.33 | 29.01 ± 4.61 | 15.25 ± 11.78 |
| | | | | Leaves | 6.78 ± 2.60 | 24.81 ± 12.75 | 12.22 ± 12.37 |
| | | | | Stems | 18.36 ± 3.48 | 22.41 ± 8.18 | 7.12 ± 13.18 |
| 2 | *Glinus oppositifolius* (L.) Aug.DC. | Molluginaceae | Fever, liver diseases, abdominal pain and jaundice treatment [16]. | Stems | 20.40 ± 7.60 | 20.61 ± 3.53 | 12.82 ± 11.14 |
| 3 | *Vernonia amygdalina* Delile | Compositae | Stomach disorder, skin wound, diarrhea, scabies, ascariasis, tonsillitis, fever and worms infection [27]. Anticancer [28]. | Leaves | 27.06 ± 12.09 | 21.82 ± 4.57 | 0.00 ± 13.42 |
| 4 | *Curcuma aromatica* Salisb. | Zingiberaceae | Bruises, corn, sprains, dysentery and gastric ailments treatment [29]. Healing wounds and fractured bones [29]. Anti-tumor [30] and anti-carcinogenic [31]. | Roots | 13.06 ± 5.66 | 19.76 ± 4.94 | 0.00 ± 3.79 |
| | | | | Leaves | 11.59 ± 3.88 | 21.00 ± 7.59 | 7.05 ± 13.41 |
| 5 | *Phyllanthus amarus* Schumach. & Thonn. | Phyllanthaceae | Cooling, diuretic, stomachic, febrifuge and antiseptic. Hepatitis, jaundice, fever, snake bites treatment [16]. | Roots | 21.48 ± 3.50 | 31.69 ± 7.33 | 17.09 ± 10.96 |
| | | | | Stems | 16.88 ± 4.17 | 20.13 ± 8.33 | 3.63 ± 6.01 |
| 6 | *Crinum* sp. | Crinum | Emetic, rheumatism, earache [32]. Swelling, urinary tract problems and anticancer [33]. | Leaves | 12.59 ± 1.96 | 15.41 ± 5.13 | 0.35 ± 4.94 |
| 7 | *Zingiber zerumbet* (L.) Roscoe ex Sm. | Zingiberaceae | Stomach pains, diarrhea, inflammation, flatulence, fever, poisoning, allergies, and bacterial infections [34]. Joint inflammation and pain [35]. Anticancer [36]. | Roots | 6.59 ± 3.43 | 18.11 ± 7.26 | 0.00 ± 7.72 |
| | | | | Leaves | 9.90 ± 3.85 | 14.18 ± 9.86 | 6.79 ± 8.37 |
| | | | | Stems | 11.21 ± 2.89 | 17.22 ± 3.53 | 4.53 ± 3.90 |
| 8 | *Hyptis suaveolens* (L.) Poit. | Lamiaceae | Headache and fever treatment, stomach pains, diarrhea, flatulence, wound healing, hemostasis, snake bites [16]. Cancers and tumors treatment [37]. | Roots | 75.25 ± 4.36 | 84.77 ± 1.23 | 73.84 ± 3.05 |
| | | | | Stems | 24.10 ± 3.67 | 40.23 ± 5.47 | 35.39 ± 6.11 |
| | | | | Leaves | 29.47 ± 4.38 | 40.05 ± 14.55 | 31.77 ± 13.74 |
| 9 | *Dicliptera chinensis* (L.) Juss. | Acanthaceae | Hepatoprotection [38]. Stomachache, enteritis and diarrhea treatment [39]. | Stems | 18.12 ± 2.75 | 22.95 ± 6.91 | 9.64 ± 4.90 |
| | | | | Leaves | 20.36 ± 5.32 | 22.01 ± 6.39 | 4.46 ± 4.56 |
| 10 | *Curcuma zedoaria* (Christm.) Roscoe | Zingiberaceae | Aid digestion, relief for colic, blood purifier [16]. Stomach diseases, leucoderma, toothache and promoting menstruation [17,40]. Anticancer [41]. | Roots | 12.00 ± 1.39 | 24.86 ± 0.57 | 11.64 ± 15.02 |
| | | | | Leaves | 24.78 ± 8.82 | 45.00 ± 13.46 | 19.50 ± 2.94 |

**Table 1.** *Cont.*

| No. | Scientific Name | Plant Family | Traditional Use | Plant Part | Cytotoxicity Effect (%) [1] | | |
|---|---|---|---|---|---|---|---|
| | | | | | Huh-7 Cells | Hep3B Cells | HepG2 Cells |
| 11 | *Eurycoma longifolia* Jack | Simaroubaceae | Fever, jaundice, cachexia, and dropsy treatment [42,43]. Antimalarial, anti-tumor, and anticancer activity [44,45]. | Stems Leaves | 26.56 ± 4.54 19.32 ± 10.10 | 29.50 ± 1.45 33.10 ± 4.35 | 21.45 ± 6.17 15.89 ± 12.97 |
| 12 | *Solanum torvum* Sw. | Solanaceae | Liver diseases, spleen enlargement, stomachache and toothache treatment [16,46]. Haemostasis and anti-inflammation [47]. | Fruits Rhizomes Leaves | 21.81 ± 2.04 24.52 ± 2.37 30.64 ± 3.67 | 39.68 ± 5.16 34.81 ± 1.71 77.39 ± 6.53 | 17.38 ± 3.85 18.99 ± 7.83 24.67 ± 2.70 |
| 13 | *Melastoma malabathricum* L. | Melastomataceae | Liver diseases, hepatitis [16]. | Fruits Stems Leaves | 11.82 ± 1.70 25.34 ± 5.72 13.91 ± 8.95 | 25.53 ± 3.74 33.67 ± 2.44 33.70 ± 9.73 | 4.40 ± 7.71 11.96 ± 5.28 6.69 ± 8.85 |
| 14 | *Luvunga scandens* (Roxb.) Buch.-Ham | Rutaceae | Ascites, cirrhosis [15]. | Leaves | 36.34 ± 11.48 | 54.30 ± 12.19 | 73.35 ± 1.21 |
| 15 | *Carallia brachiata* (Lour.) Merr. | Rhizophoraceae | Inflammation of throat and mouth [15]. Liver diseases (folk experience). | Rhizomes Leaves | 26.11 ± 6.18 16.57 ± 9.42 | 25.73 ± 2.08 31.06 ± 0.31 | 12.37 ± 3.03 12.43 ± 14.15 |
| 16 | *Helicteres hirsuta* Lour. | Malvaceae | Ulcer, anti-malaria, dysentery, flu, smallpox [14]. Liver diseases (folk experience) [48]. | Leaves | 23.08 ± 9.32 | 28.47 ± 12.11 | 12.93 ± 8.18 |
| 17 | *Miliusa velutina (A.DC.)* Hook.f. & Thomson | Annonaceae | Eye inflammation, stomach pains, sinusitis, skin diseases [15]. Liver diseases (folk experience). | Stems Leaves | 14.22 ± 5.61 15.58 ± 9.08 | 21.93 ± 4.97 31.22 ± 12.21 | 0.10 ± 2.27 7.02 ± 2.41 |
| 18 | *Grewia nervosa* (Lour.) Panigrahi | Malvaceae | Cough, anti-malaria, digestive disorders [15]. Liver diseases (folk experience). | Fruits Stems Leaves | 14.82 ± 5.63 31.00 ± 6.84 12.86 ± 4.09 | 24.85 ± 5.79 45.50 ± 1.04 20.18 ± 9.89 | 7.95 ± 4.64 6.73 ± 12.07 4.17 ± 1.78 |
| 19 | *Oroxylum indicum* (L.) Kurz | Bignoniaceae | Cough, ulcer, abdominal pain [17]. Liver diseases (folk experience). | Stems Leaves | 33.90 ± 7.97 14.80 ± 8.18 | 32.36 ± 10.17 25.65 ± 10.84 | 23.80 ± 12.57 7.17 ± 9.15 |
| 20 | *Cleome gynandra* L. | Cleomaceae | Arthritis, pimples, fever, anti-malaria, dysentery [16]. Liver diseases (folk experience). | Stems Leaves | 34.55 ± 7.12 17.37 ± 6.44 | 22.76 ± 8.05 25.80 ± 2.46 | 11.11 ± 3.60 8.96 ± 4.57 |

**Table 1.** *Cont.*

| No. | Scientific Name | Plant Family | Traditional Use | Plant Part | Cytotoxicity Effect (%) [1] | | |
|---|---|---|---|---|---|---|---|
| | | | | | Huh-7 Cells | Hep3B Cells | HepG2 Cells |
| 21 | *Lasia spinosa* (L.) Thwaites | Araceae | Liver diseases, hepatitis, ascites, cirrhosis [15]. Liver failure, liver tonic [16]. | Fruits | 23.48 ± 12.26 | 31.81 ± 10.98 | 2.71 ± 6.98 |
| | | | | Rhizomes | 13.82 ± 11.69 | 38.22 ± 12.09 | 12.64 ± 4.24 |
| | | | | Stems | 18.17 ± 7.59 | 26.31 ± 8.87 | 4.65 ± 6.69 |
| | | | | Leaves | 7.68 ± 10.18 | 25.06 ± 6.46 | 7.75 ± 12.37 |
| 22 | *Ipomoea pes-tigridis* L. | Convolvulaceae | Pimples, hemoptysis [15]. Liver diseases (folk experience). | Roots | 29.25 ± 10.95 | 18.24 ± 8.19 | 7.11 ± 6.30 |
| | | | | Stems | 14.97 ± 6.72 | 28.99 ± 9.28 | 14.32 ± 11.86 |
| | | | | Leaves | 11.42 ± 6.41 | 24.15 ± 3.43 | 13.17 ± 8.12 |
| 23 | *Cleome rutidosperma* DC. | Cleomaceae | Hepatitis (folk experience). | Stems | 11.97 ± 8.95 | 27.19 ± 7.27 | 1.06 ± 9.51 |
| | | | | Leaves | 29.47 ± 1.37 | 24.64 ± 3.64 | 18.61 ± 6.58 |
| 24 | *Crotalaria pallida* Aiton | Leguminosae | Anticancer [16]. | Stems | 23.80 ± 5.87 | 16.82 ± 5.59 | 11.64 ± 5.77 |
| | | | | Leaves | 10.01 ± 7.22 | 21.80 ± 6.98 | 15.84 ± 4.57 |

[1] Cytotoxicity effect (%) indicates the mean maximum cytotoxic effect on tumor cells at a concentration of 100 µg/mL. All data were calculated from three independent tests (mean ± SD).

*3.4. Evaluation of TFC, Antioxidant Activity, and $CC_{50}$ of the Extracts from Solanum torvum Leaves*

The total flavonoids content, the antioxidant activity, and the cytotoxicity effect of the extracts from *S. torvum* leaves were evaluated and designated in Figure 4 and Tables S1–S3. The highest TFC value was found in *n*-hexane extract (230.64 ± 16.74 mg RE/g extract).

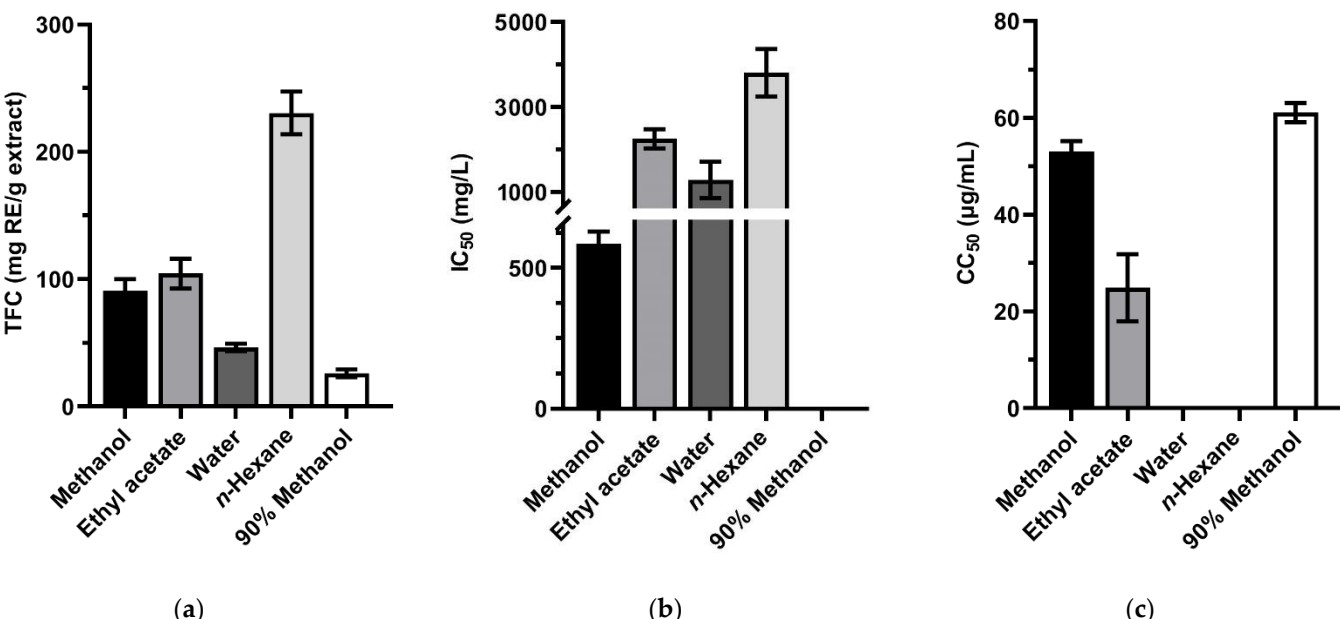

(**a**)  (**b**)  (**c**)

**Figure 4.** (**a**) Total flavonoids content, (**b**) the antioxidant activity in DPPH assay and (**c**) the cytotoxicity effect on Huh-7 HCC cells of *Solanum torvum* leaves extracting in methanol, ethyl acetate, water, *n*-hexane, and 90% methanol. For panel (**b**), the 90% methanol bar is unseen because the extract did not exhibit the antioxidant capacity. No bar in panel (**c**) is shown for the extract that had a 50% cytotoxic concentration of more than 100 μg/mL. Values are presented as mean ± SD (*n* = 3). Error bars show standard deviation.

The scavenging effects of all the extracts on DPPH radicals did not show significant antioxidant capacity with $IC_{50}$ values > 250 μg/mL, and the 90% methanol extract was not determined scavenging activity hidden in Figure 4b [49].

Results from the cytotoxicity assay of *S. torvum* leaves showed a strong cytotoxicity effect with the $CC_{50}$ was 24.9 ± 6.9 μg/mL in ethyl acetate extract as well as moderate cytotoxicity effect in the case of the methanol and 90% methanol extracts (53 ± 2.2, 61.1 ± 2.0 μg/mL, respectively) on Huh-7 HCC cells.

## 4. Discussion

*4.1. Total Flavonoids Content, Antioxidant Activity, and Cytotoxic Effect of the Extracts from Luvunga scandens Leaves*

The study by Permanasari et al. reported that the *n*-hexane and dichloromethane extract from *L. scandens* leaves at a concentration of 30 μg/mL showed the ability to inhibit 97.88 ± 0.80%, and 83.97 ± 1.02% the growth Huh-7 HCC cells, respectively [50]. In this study, the *n*-hexane extract of *L. scandens* leaves did not show cytotoxic effect, while the ethyl acetate extract and methanol extract showed moderate cytotoxicity ability with $CC_{50}$ values were 36.5 ± 2.9 and 38.5 ± 2.1 μg/mL, respectively. The high TFC value of ethyl acetate extract (233.46 ± 17.34 mg RE/g extract) could be due to the partial extraction procedure by which the compounds having cytotoxic effects were more concentrated in the ethyl acetate extract.

*4.2. Total Flavonoids Content, Antioxidant Activity, and Cytotoxic Effect of the Extracts from Hyptis suaveolens Roots*

The extracts from *H. suaveolens* roots exhibited the superior cytotoxic potential in Huh-7 HCC cells with the $CC_{50}$ value ranging from 1.6 to 14.4 µg/mg. Although the roots of *H. suaveolens* have been used as Vietnamese traditional medicine plants to treat liver cancer for years, the cytotoxic effect on HCC cell lines has never been reported. Nevertheless, numerous cytotoxic activity studies on the *H. suaveolens* extracts against other cancer cell lines have been published. Indeed, the ethanol extract from *H. suaveolens* twigs, leaves, and flowers had been studied for its strong anticancer potent in the KB cell culture and the P-388 lymphocytic leukemia system for a long time [51]. Gurunagarajan and Pemaiah reported that the ethanol extract from the aerial parts of *H. suaveolens* had strong activity against Ehrlich Ascites carcinoma cell lines with the $CC_{50}$ was of 10.63 µg/mL [52]. In addition, *H. suaveolens* leaves were proposed to protect the liver against $CCl_4$-induced oxidative damage in rats. Furthermore, the water extract from *H. suaveolens* leaves was also reported to have hepatoprotective potentials on acetaminophen-induced liver damage in rabbits [53]. Our finding suggested for the first time that the roots of *H. suaveolens* also possess anti-HCC activities, and it is also worth evaluating its liver protective effect.

Among three herbal medicines in this study, the antioxidant activity of *H. suaveolens* root extracts, except for the *n*-hexane extract, were the most prominent with $IC_{50}$ values ranging from 12.59 to 29.67 µg/mL. In a previous study, the methanol extract from *H. suaveolens* leaves has been reported to have a strong antioxidant activity ($IC_{50}$ = 24.56 µg/mL) [54]. Another study on the extracts from *H. suaveolens* leaves in polar solvents supported that they had remarkable antioxidant activity; particularly, the 70:30 ethanol/water extract possessed a maximum antioxidant activity with $IC_{50}$ = 2.73 ± 0.005 µg/mL [55]. Until now, there have been no studies on evaluating the antioxidant capacity from the extract of *H. suaveolens* roots. In this study, the cytotoxic effect and antioxidant capacity of *H. suaveolens* roots are directly proportional with the high values of total flavonoids content in the ethyl acetate and 90% methanol extracts.

*4.3. Total Flavonoids Content, Antioxidant Activity, and Cytotoxic Effect of the Extracts from Solanum torvum Leaves*

Although the cytotoxicity effect screening results of 24 medicinal herbs did not show the notable cytotoxicity of the crude methanol extract from *S. torvum* leaves against HCC Huh-7 HCC cells (Table 1), the ethyl acetate partial extract from *S. torvum* leaves exhibited a potent cytotoxicity against Huh-7 HCC cells with $CC_{50}$ of 24.9 ± 6.9 µg/mL (Table S3). A previous study also reported that the ethyl acetate extract from *S. torvum* fruits displayed better cytotoxic capacity against MCF-7 cells than its methanol extract and *n*-hexane extract [56].

Results in our study showed that the partial extracts from *S. torvum* had no antioxidant activity. As shown in Figure 4a and Table S1, the TFC value of extracts from *S. torvum* leaves ranged from 23.13 to 247.38 mg RE/g extract, which are in good agreement with previous studies. Indeed, Waghulde et al. reported that the total flavonoids content of *S. torvum* ethanol extract and methanol extract were 85.26 ± 0.02 and 65.69 ± 0.02 mg RE/g extract, respectively [57]. Another study was carried out on *S. torvum* fruits using Soxhlet extraction with petroleum ether, chloroform, ethyl acetate, methanol, and water showed that the TFC value of ethyl acetate, methanol, and water extracts were 1.013, 31.043, and 13.064 mg RE/g extract, respectively while the extracts in petroleum ether and chloroform did not present flavonoid [58].

## 5. Conclusions

This study screened 52 crude extracts from different parts of 24 traditional medicinal herbs for liver cancer treatment in Vietnam with outstanding cytotoxic activity against three HCC cell lines. The crude methanol extracts of *L. scandens* leaves, *H. suaveolens* roots, and *S. torvum* leaves were identified with potent anti-HCC activity. Then, we obtained four

partial extracts with different polarity solvents, including ethyl acetate, water, *n*-hexane, and 90% methanol, and investigated the total flavonoids content, free radical scavenging activity, and cytotoxicity effect of all extracts. The methanol, ethyl acetate, and 90% methanol extracts of *H. suaveolens* roots and the ethyl acetate extract of the leaves of *L. scandens* and *S. torvum* exhibited strong cytotoxicity against Huh-7 HCC cells. These extracts could be promising sources for the discovery of novel therapeutic modalities for HCC treatment. Further studies are needed to isolate, identify, and quantify the bioactive constituents responsible for the anti-HCC capacity in each recommended extract.

**Supplementary Materials:** The following are available online at https://www.mdpi.com/article/10.3390/pr9111956/s1; Table S1. Total flavonoids content (TFC) values of methanol, ethyl acetate, water, *n*-hexane, and 90% methanol extracts from *Luvunga scandens* leaves, *Hyptis suaveolens* roots, and *Solanum torvum* leaves were determined by using the aluminum-flavonoid complex colorimetric method: Table S2. The antioxidant activity $IC_{50}$ values of methanol, ethyl acetate, water, *n*-hexane, and 90% methanol extracts from *Luvunga scandens* leaves, *Hyptis suaveolens* roots, and *Solanum torvum* leaves were determined by using DPPH assay; Table S3. The cytotoxicity effect on Huh-7 cells $CC_{50}$ values of methanol, ethyl acetate, water, *n*-hexane, and 90% methanol extracts from *Luvunga scandens* leaves, *Hyptis suaveolens* roots, and *Solanum torvum* leaves were determined by using resazurin assay.

**Author Contributions:** All authors contributed to designing and conducting experiments, data analysis, and interpretation as well as drafting and revising the manuscript. All authors have read and agreed to the published version of the manuscript.

**Funding:** This research is funded by Vietnam National University HoChiMinh City (VNU-HCM) under grant number C2019-44-02.

**Acknowledgments:** We thank the staff from the National Natural Product Libraries and High-Throughput Screening Core Facility (NPS core lab) at Kaohsiung Medical University for technical assistance.

**Conflicts of Interest:** The authors declare no conflict of interest.

## Abbreviations

| | |
|---|---|
| HCC | Hepatocellular carcinoma. |
| MTT | 3-(4,5-Dimethylthiazol-2-yl)-2,5-diphenyltetrazolium bromide. |
| TFC | Total flavonoids content. |
| Huh-7 HCC cells | Huh-7 human hepatocellular carcinoma cells. |
| DPPH | 1,1-Diphenyl-2-picrylhydrazyl. |
| $IC_{50}$ | The half maximal inhibitory concentration. |
| $CC_{50}$ | The 50% cytotoxic concentration. |
| DMEM | Dulbecco's modified Eagle's medium. |
| RE | Rutin equivalent. |
| *L. scandens* leaves | *Luvunga scandens* (Roxb.) Buch.-Ham. ex Wight & Arn. leaves. |
| *H. suaveolens* roots | *Hyptis suaveolens* (L.) Poit. roots. |
| *S. torvum* leaves | *Solanum torvum* Sw. leaves. |

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
