# Peer review of "Bioassay-Guided Discovery of Potential Partial Extracts with Cytotoxic Effects on Liver Cancer Cells from Vietnamese Medicinal Herbs"

_processes, doi:10.3390/pr9111956_

Round 1

Reviewer 1 Report

  1. The author should address traditional uses of each herbs selected for the study.
    2. The author should state the difference of Huh7, Hep3B, and HepG2 cell lines in terms of their uniqueness of these cells in pathology, drug-resistance in order to explain why the author studied in these cells.
    3. There is a misspelling of L. scandens in the abstract.
    4. How about the cytotoxicity of promising extracts towards normal cells?

Author Response

Firstly, we would like to thank the reviewers for their thorough assessment and constructive remarks. We believe that these comments have contributed to significant and clarified concepts made in our work.

Please see enclosed our revised manuscript. We have addressed all the concerns raised by the reviewers.

1.1 The author should address traditional uses of each herb selected for the study?

Thanks for the suggestion. We have already supplemented a column of the traditional uses of each herb selected in Table 1. Please see the present revised manuscript.

1.2 The author should state the difference of Huh7, Hep3B, and HepG2 cell lines in terms of the uniqueness of these cells in pathology, drug-resistance in order to explain why the author studied in these cells?

Thanks for the comment. Studies on cytotoxic activity against liver cancer cell lines have been carried on numerous different cell lines. The choice of what hepatoma cell line to study is highly variable, and there is no final rule to guide the decision; thus, the selection of hepatoma cell lines could depend on the aspect of cancer under investigation. In previous studies, HepG2 and Huh-7 cell lines were reported for the studies requiring the presence of hepatitis C virus replicons, while HepG2 and Hep3B cells were used for studies requiring the presence of hepatitis B virus. In addition, HepG2 and Huh-7 cell lines were reported to maintain easily in culture and retain several hepatocyte functions. As well as some advantages such as the relatively low cost, high-throughput capacity, and great availability, we decided to study the cytotoxicity effect on Huh-7, Hep3B, and HepG2 cell lines.

1.3 There is a misspelling of L. scandens in the abstract.

We thank the reviewer for carefully reading our manuscript. We have already replaced “L. scandenss” with “L. scandens” in the abstract section in the revised version of this manuscript.

1.4 How about the cytotoxicity of promising extracts towards normal cells?

Thanks for raising the question. In this study, we did not perform the cytotoxicity test of the promising extracts towards normal cells. We desired to focus on pointing out and clarifying the cytotoxic activity on Huh-7 HCC cells of the extracts from the three prospective medicine herbs. In addition, those medicinal herbs have been used for a long time for the traditional treatment of liver diseases with no side effects reported in folk documents.

Reviewer 2 Report

The manuscript entitled "Evaluation of cytotoxicity, antioxidant properties and total flavonoids contents of partial extracts from three traditional herbal plants" described the physical-chemical characterization and in vitro cytotoxicity of 24 traditional Vietnamese medicinal plants, emphasizing their  anti-HCC potential.

The paper cannot be published in this form; the material and methods are poorly described and the results are not convergent. Therefore I suggest to improve and resubmit the manuscript.

The first observation is that the paper does not correspond to the Aims and Scope of the Processes MDPI Journal, since it lacks exactly the elements which could convey this study towards original biologic processes, novel technologies for the plant material processing, description of novel excipients in pharmaceutical processes, original natural and herbal remedies processing, or other aspects. At least for the three selected -original and novel- extracts, technology details and methodological novelties should be featured, and shared by the authors.  

Regarding the plants origin, a more exact geographical localization (longitude/latitude) and the exact period of harvesting should be inserted, or literature reference if available; they are no details on the plants biologic categorization; harvesting methods and drying methods are shortly described. No Supplementary Materials were available in this revision process ( also no such reference in the texts). Certainly, there would have been many details of interest for the Processes MDPI readers in this chapter.

Preparation of extracts, chemical analysis of their content and the redox potential should be described prior to the cytotoxicity testing, and the elements which confer originality to the methods and processes should be highlighted to fit to the Journals topic.

A major concern is that the MTT assay-derived IC50 values were not expressed quantitatively; only one concentration, that of 100 micrograms/mL is not sufficient to serve as basis for a screening. Also in MTT, and resazurin-based experiments authors should highlight the novel elements related to the 24 plant extracts cell-based analyze; these aspects are very important from the viewpoint of the reproducibility.  

Figure 2 contains inexact information, 2b represent no DPPH results; it is not clear if the resazurin-based assay was performed on a single concentration similar to MTT (in this case it is insufficient biologic data).      

Complete data regarding the flavonoid content should be given in a table.

Which cell bank provided Huh7, Hep3B and HepG2 cells? No data was given regarding the cell culture media. How many microliters of cell suspension were in a well on the 96-well plate and what was the final concentration of the DMSO in the cell media?

They are minor observations such:

Introduction line 45, the phrase about chemotherapy in HCC should be reconsidered since the reference is from 2014 and meanwhile many therapeutic advances were done in HCC chemotherapy.

In some places the phrases should be rewritten, an example: abstract row 23: "are the good candidates against..." and please explain in the abstract on what basis the three extracts were selected.

Chapter 6: this chapter should contain only information on patents if any, no other data.

Author Response

Firstly, we would like to thank the reviewers for their thorough assessment and constructive remarks. We believe that these comments have contributed to significant and clarified concepts made in our work.

Please see the attachment. We have addressed all the concerns raised by the reviewer.

Reviewer 3 Report

 Evaluation of cytotoxicity, antioxidant properties and total flavanoids contents of partial extracts from three traditional 3 herbal plants by

 Hien Nguyen Minh et al

 Abstract :  The author should give a clarity on the selection of plants used as well as the solvents used. Explain the rationale behind the selection of  ethyl acetate and 90% methanol extract. The sentence like “24 medicinal herbs traditionally used to treat liver cancer in Vietnam” should be included in the abstract , then the concept will be more clear.

 “Chemotherapy 45 is not regularly performed in the HCC treatment due to the toxicity and serious side effects” The sentence need to be rewritten

Introduction. The rationale for plant derived compounds against HCC is not clear. The part should be rewritten as  the rationale for choosing the plants, how the flavanoid component is more relevant etc.

Methods: Huh7 HCC cells can be rewritten as Huh-7

  The time period need to be fixed as 48/72h

It is better to arrange the draft content as  antioxidant, total phenolic content, cytotoxicity etc.

 It is better to rewrite the title more interestingly.

 I suggest major revision. The authors should consult with any cancer biology gp for the clear interpretation of the biological activity

Author Response

Firstly, we would like to thank the reviewers for their thorough assessment and constructive remarks. We believe that these comments have contributed to significant and clarified concepts made in our work.

Please see the attachment. We have addressed all the concerns raised by the reviewers.

Round 2

Reviewer 2 Report

The manuscript of H.N. Minh and co-authors was improved, starting from the title and introductory part; a supplementary file was attached and data were inserted also to Table 1, methods, results, references and other essential parts of the manuscript. 

Since the authors computed the IC50 values for DPPH test, this refer to the free radical scavenging only. The cytotoxicity test was made for only one concentration,  it is actually a percent of cell growth inhibition for a single concentration of 100 μg/mL solution.  It is not described the quantity of DMSO added to cell culture media for each well, for example: 10 microliters of extract in DMSO to 90 microliter media, or to 190 microliters media. The inhibitory effect of the solvent have to be given, to prove that the growth inhibition percents provided in Table 1 are due to the extract and not the solvent, DMSO in this case.  The term "cytotoxicity" is more appropriate for the extracts effect against the Huh-7 cells, where resazurin-based tests were done for several concentrations, for 72-hours interval. 

In the Conclusion section, what methods and processes are recommended by the authors in the further steps towards potential pharmaceutical  use? They were described extremely valuable information about a series of 24 plants: some of them were proven to have no activity against the hepatocarcinoma cells, but certainly they could be very useful in other cancers or benign diseases. What is the recommendation of the authors for further biologic testing? 

Author Response

Again, we would like to thank the reviewer for your thorough assessment and constructive remarks. We believe that these comments have contributed to significant and clarified concepts made in our work.

Please see the attachment. We have addressed all the concerns raised by the reviewer.

Reviewer 3 Report

 The authors can check the toxicity of the selected extracts/extract on any of the normal cell line to confirm the potent extract/fraction is safe

Author Response

Firstly, we would like to thank the reviewers for their thorough assessment and constructive remarks. We believe that these comments have contributed to significant and clarified concepts made in our work.

Please see enclosed our revised manuscript. We have addressed all the concerns raised by the reviewer.

The authors can check the toxicity of the selected extracts/extract on any of the normal cell line to confirm the potent extract/fraction is safe.

Thanks for the raising suggestion. Those medicinal herbs have been used for a long time for the traditional treatment of liver diseases with no side effects reported in folk documents. Therefore, we did not perform the cytotoxicity test of the promising extracts towards normal cells.